# Harnessing ctDNA in Advanced Melanoma: A Promising Tool for Informed Clinical Decisions

**DOI:** 10.3390/cancers16061197

**Published:** 2024-03-18

**Authors:** Rugile Pikturniene, Alvydas Cesas, Sonata Jarmalaite, Arturas Razbadauskas, Vincas Urbonas

**Affiliations:** 1Life Sciences Center, Institute of Biosciences, Vilnius University, 7 Sauletekio Ave, 01513 Vilnius, Lithuania; rugile.pikturniene@kulig.lt (R.P.); sonata.jarmalaite@nvi.lt (S.J.); 2Chemotherapy Unit, Department of Oncology, Klaipeda University Hospital, Liepojos 49, 92288 Klaipeda, Lithuania; alvydas.cesas@kulig.lt (A.C.); rektorius@ku.lt (A.R.); 3Laboratory of Clinical Oncology, National Cancer Institute, Baublio 3B, 08406 Vilnius, Lithuania; 4Faculty of Health and Sciences, Klaipeda University, H. Manto 84, 92294 Klaipeda, Lithuania

**Keywords:** melanoma, ctDNA, biomarkers, immunotherapy

## Abstract

**Simple Summary:**

Cutaneous melanoma, a fatal and aggressive tumor, has witnessed a transformative shift in its clinical management over the past decade with the advent of anti-programmed cell death 1 (PD1) and anti-cytotoxic-T-lymphocyte-associated antigen 4 (CTLA-4) immunotherapies, as well as targeted therapies. Current standard monitoring methods, such as imaging scans, have limitations, necessitating the exploration of alternative biomarkers. Circulating tumor DNA emerges as a promising blood-based biomarker for precise clinical decisions.

**Abstract:**

Cutaneous melanoma, an aggressive malignancy, has undergone significant transformation in clinical management with the introduction of immune checkpoint inhibitors (ICIs) and targeted therapies. Current monitoring methods, such as imaging scans, present limitations, prompting exploration of alternative biomarkers. This review comprehensively explores the role of circulating tumor DNA (ctDNA) in advanced melanoma, covering technical aspects, detection methods, and its prognostic and predictive value. Recent findings underscore ctDNA’s potential applications and implications in clinical practice. This review emphasizes the need for precise and dynamic biomarkers in melanoma care, positioning ctDNA as a promising blood-based tool for prognosis, treatment response, and resistance mechanisms. The technical nuances of ctDNA detection, association with melanoma mutations, and its role in guiding therapeutic decisions for immunotherapy and targeted therapy underscore its multifaceted utility, marking a paradigm shift in clinical decision-making and offering a promising trajectory for personalized and informed care in advanced melanoma.

## 1. Introduction

The incidence of cutaneous melanoma is increasing, with an estimated 325,000 cases reported globally per year [1]. Melanoma is a fatal and aggressive tumor, as it is associated with poor prognosis in patients with advanced or metastatic disease and it causes most of the skin cancer-related deaths [2]. Over the last decade, there has been a revolutionary transformation in the clinical treatment of individuals with advanced melanoma, brought about by the adoption of immunotherapies targeting PD1 and/or CTLA-4, along with targeted therapies inhibiting BRAF/MEK [3]. Imaging scans, such as ^18^F-labelled fluorodeoxyglucose positron emission tomography (FDG-PET) or computed tomography (CT), are currently the clinical standard for treatment monitoring, but are costly and of limited accessibility in some areas [4]. Nevertheless, a considerable proportion of patients either develop acquired resistance or do not experience therapeutic benefits, and in certain instances, treatment can pose toxicity risks, potentially leading to fatalities [5]. Consequently, closely monitoring disease progression and assessing patient prognosis are crucial, playing a pivotal role in enhancing the quality of life of individuals with melanoma.

Presently, the guidelines provided by the National Comprehensive Cancer Network (NCCN) suggest periodic imaging and/or clinical assessments to evaluate treatment effectiveness or identify disease progression [6]. Nonetheless, the frequency at which imaging-based surveillance can be conducted is constrained. This approach has its drawbacks, including the potential for false positives and result misinterpretation, which may result in costly and sometimes unnecessary procedures [7].

While plasma lactate dehydrogenase (LDH) levels can serve as a prognostic indicator in advanced melanoma, only 30–40% of stage IV patients exhibit elevated LDH at the outset. Moreover, such elevation can often result from treatment toxicity or other factors unrelated to the disease, such as tissue or organ damage, pregnancy-related complications, and certain autoimmune diseases [8]. Elevated levels of S100 and C-reactive protein are linked to the presence of detectable ctDNA, while LDH levels in many studies did not demonstrate the same correlation [9]. However, at present, there are no other blood-based biomarkers with clinical utility for the real-time assessment of treatment response or disease progression, hindering the optimization of subsequent treatment strategies.

This review will explore the significance of circulating tumor DNA (ctDNA) as a non-invasive “real-time” biomarker, capable of offering diagnostic and prognostic insights prior to treatment initiation, during the course of treatment, and upon disease progression (Figure 1).

This review was performed in accordance with the PRISMA (Preferred Reporting Items for Systematic Reviews and Meta-Analyses) guidelines and has not been registered.

## 2. ctDNA as a Blood-Based Biomarker

Although the utilization of circulating cell-free DNA (cfDNA) in clinical applications is a relatively recent development, the presence of cfDNA was initially documented in 1948 [10]. CfDNA pertains to the unencapsulated, freely circulating DNA molecules released or shed from cells. This phenomenon is a normal occurrence in healthy tissues, attributed to cell turnover and intentional shedding into the circulation [11].

The function of cfDNA is not clearly understood, and it could potentially serve as a cellular waste product that is eventually eliminated from circulation through the kidneys as urine cfDNA. Furthermore, the short half-life of cfDNA, estimated to be 1 to 2 h [12], is primarily attributed to blood DNases that facilitate rapid DNA degradation [13].

Recent studies have demonstrated the potential of plasma ctDNA as an emerging blood-based biomarker, acting as prognostic and predictive tool in melanoma. In the bloodstream of healthy individuals, cfDNA is present at lower concentrations, whereas in patients with cancer, it is found at significantly higher levels [14]. The mechanisms underlying the release of cfDNA are not fully comprehended; nevertheless, it is hypothesized to result from cell necrosis, apoptosis, and the secretion of cells phagocytosed by macrophages [15].

CtDNA is derived from cancer cells, adding to the overall cfDNA in the bloodstream and consequently elevating the concentration of cfDNA in the presence of cancer [16].

Blood, particularly plasma, serves as the optimal substance for collecting ctDNA and is primarily employed in clinical settings for companion diagnostic applications. This involves identifying mutations associated with some approved targeted therapies. While plasma is the favored sample for cfDNA analyses, it is worth noting that cfDNA can also be identified in serum. However, it is important to recognize that the integrity of cfDNA in serum is lower compared to that in plasma [17,18,19]. Nevertheless, alternative sources/origins of cfDNA and ctDNA have been scrutinized, such as cerebrospinal fluid [20] and urine ctDNA [21]. In the context of prevailing commercial and research assays, the standard procedure involves the isolation of cfDNA by initially separating plasma from whole blood through double centrifugation protocols, aimed at thorough cell removal. Subsequently, nucleic acids are extracted using diverse techniques [22]. It is highly essential to emphasize the careful handling of cfDNA, necessitating the utilization of dedicated equipment such as centrifuges, pipettes, and the provision of controlled environments like “clean rooms” and dedicated hoods. These precautions are vital to prevent sample contamination with aerosolized DNA, especially in the context of NGS (Next-Generation Sequencing) and other assays capable of detecting mutations at extremely low levels, such as 0.01% Variant Allele Frequency (VAF) [23].

In general, a minimum of 15–20 mL of whole blood is recommended for optimal plasma DNA yield, although this requirement may vary based on the characteristics of the assay employed [24]. It is optimal to isolate cfDNA within a few hours of blood collection to avoid white blood cell lysis. Delayed processing can lead to the release of substantial amounts of cellular genomic DNA into the plasma component within the blood collection tube [25,26]. Extended clotting and delayed separation of plasma from blood cells over a 24-h period significantly increase both the concentration and observed size of cell-free DNA in blood samples. Additionally, repeated cycles of freezing and thawing of plasma samples, but not extracted DNA, results in DNA fragmentation. As a consequence, it is crucial to handle blood samples intended for the examination of ctDNA integrity within a 6-h timeframe after collection. The plasma should be divided into smaller portions to prevent repeated freezing and thawing. For storage purposes, extracting DNA from the plasma samples is recommended because DNA tends to exhibit greater resistance to fragmentation when stored in a DNA extraction solution compared to being stored directly in plasma [17]. In individuals without health issues, cfDNA concentrations typically fall within the range of 0.1 to 15 ng/mL of blood. These levels tend to increase following activities such as exercise, tissue injury, and inflammation [27]. Another study, published in 2023 by Lauren G. Aoude and colleagues [28], stated that the median ctDNA concentration for patients with advanced melanoma was 5.95 ng/mL, ranging from 1.90 to 141.53 ng/mL. To assess the association between ctDNA concentration and survival, a univariable survival analysis using a log-rank test (Mantel–Cox) was conducted. A cutoff of 10 ng/mL was applied. Patients diagnosed with stage IV melanoma, displaying ctDNA concentrations exceeding 10 ng/mL, exhibited significantly diminished Disease-Specific Survival (DSS) (median, 6.82 months; range, 1.35 to 26.83 months; *p* < 0.0001) and progression-free survival (PFS) (median, 4.98 months; range, 1 to 22 months; *p* = 0.0015) in comparison to those with ctDNA concentrations below 10 ng/mL (DSS: median, 42.85 months; range, 0.56 to 60 months; PFS: median, 20.20 months; range, 0.56 to 60 months).

The importance of having undetectable ctDNA at the initial assessment remains noteworthy even in multivariable analyses incorporating factors like LDH, CRP, the presence of more than three metastatic sites, and performance status. Interestingly, a recent study [29] proposes that baseline ctDNA functions as a primary biomarker for treatment response, especially in the context of first-line treatment.

This underscores the evolving understanding of the role of ctDNA in predicting treatment response and emphasizes the importance of context, such as treatment line, in interpreting its prognostic value.

ctDNA can be used in patients who are unable to undergo surgery or excisional tissue collection, or in cases where tumors are difficult to access. In theory, the genomic alterations identified through liquid biopsy reflect the primary tumor burden, while tissue samples offer site-specific information [30].

There are various techniques for analyzing ctDNA, spanning from digital methods that target specific point mutations to targeted panels that usually examine around 1000 genes. Broader approaches include whole exome sequencing (WES) and assays based on methylation [31].

Nonetheless, it is essential to underscore that selecting the appropriate assay for different clinical scenarios is pivotal when contemplating the integration of ctDNA into clinical practice [32].

In the context of creating an assay for detecting minimal residual disease (MRD), a comprehensive yet shallow whole exome sequencing might not possess the required sensitivity for discerning micro-metastatic disease. Nevertheless, this method could prove more valuable in uncovering new resistance mechanisms to therapy. While the druggable gene mutations must be studied in a ctDNA-based companion diagnostic test, early epigenetic changes in cfDNA might be a more relevant target for liquid biopsy-based diagnostic testing.

Every assay has its own array of strengths and weaknesses, impacting its appropriateness for a given situation. Innovative approaches like MRD-EDGE employ machine learning-driven denoising and an extended feature space that encompasses fragmentomics and the allelic frequency of germline single-nucleotide polymorphisms, thereby challenging the traditional paradigm [33].

When dealing with ctDNA, clinicians should carefully consider the specific information crucial for the patient’s current disease stage to address the clinical question effectively. Customizing the ctDNA assay based on the priorities of the patient’s condition is paramount. Moreover, if ctDNA transitions from experimental settings to clinical application, clinicians need a profound understanding of the test performance within their patient population. They should also be aware of its limitations when making treatment decisions based on the obtained results [34].

Furthermore, in certain cancers, identifying driver mutations like BRAF V600 may be relatively straightforward, while in others, such as cases requiring multi-regional sequencing, identifying truncal mutations might be necessary. This variance affects the feasibility of selecting single-point mutations to evaluate tumor burden. Practical considerations in this context include the ease of sample collection and initial laboratory processing, the need for bioinformatics support in result interpretation, and the overall cost of conducting assays, particularly if longitudinal monitoring is part of the proposed strategy [35].

## 3. Detection of ctDNA in Locally Advanced Melanoma: Early Recurrence Monitoring and Determination of Minimal Residual Disease

Most melanomas are detected at an early stage (stage I), typically treatable through surgical removal, leading to a 97–99% 5-year survival rate. Regrettably, certain early-stage melanomas might experience recurrence after excision and progress to metastasis. Consequently, melanomas within the same stage can exhibit differences in progression and patient survival, likely attributed to undetected tumor heterogeneity upon histopathology [36].

Those with positive sentinel lymph nodes (SLNs) have a wider array of treatment choices, encompassing surgery, systemic treatment, and imaging modalities like PET and CT scans to evaluate the scope of distal metastatic spread. Routine imaging is frequently utilized to track tumor volume, location, and the efficacy of systemic therapies [37].

Research on ctDNA in melanoma has primarily concentrated on identifying relapses or responses in patients with metastatic melanoma experiencing systemic anticancer treatment. During adjuvant treatment, ctDNA has been observed to be associated with melanoma-specific survival in melanoma (stage III) and tumor burden before operation [38].

Furthermore, the detection of ctDNA post-surgery was linked to poorer relapse-free survival, increased size of melanoma deposits in lymph nodes, a higher count of lymph nodes involved in melanoma, a more advanced stage, and elevated levels of LDH among stage III melanoma patients [39]. Significantly, relapse occurrence showed a correlation with a ctDNA increase in 48% of cases, compared to 33% of patients with negative ctDNA after operation [40]. In a prospective study led by A. Forschner and colleagues, five female patients with stage IIIC or IIID melanoma were enrolled in adjuvant nivolumab therapy. Among the patients, those with radiologically evident distant metastasis showed an increase in ctDNA. On the other hand, the patients experiencing only local relapses did not exhibit detectable levels of ctDNA [41]. These findings align with the results reported by Wong and colleagues, indicating that subcutaneous disease sites are not adequately represented in plasma samples [42].

Although certain small-scale studies have demonstrated encouraging outcomes in monitoring patients with ctDNA in early stages, the overall sensitivity is relatively limited. This is likely due to the presence of more uniform tumor populations and modest concentrations of ctDNA [43].

## 4. Detection of ctDNA in Metastatic Melanoma: Treatment Guide for Patients Receiving Immunotherapy and Targeted Therapy

In the context of stage IV disease, ctDNA is a valuable tool for guiding treatment decisions and monitoring treatment response. Several studies conducted in advanced disease settings have indicated that a reduction in plasma ctDNA levels is associated with treatment response and may frequently precede radiological signs of disease progression. Moreover, the measurement of ctDNA levels proved to be a more reliable indicator of treatment response and the development of treatment resistance compared to LDH levels. Although LDH levels did exhibit changes over time, they were slower to reflect alterations in disease status and lacked accuracy [44]. Levels of ctDNA have demonstrated correlation with tumor burden observed on CT scans, and ctDNA levels have been found to correspond with the metabolic burden of the disease, as evaluated through FDG-PET scans [45].

It is worth noting, however, that the release of ctDNA may vary depending on the disease site. Patients who have involvement of bones, visceral organs, or lymph nodes often display elevated levels of ctDNA. Interestingly, these levels may not correspond with the metabolic disease burden evaluated through FDG-PET. Conversely, individuals with metastases in the brain or significant subcutaneous disease consistently demonstrate low levels of ctDNA, even when measurable disease is present [46]. The majority of patients with detectable ctDNA exhibit noticeable metastases in visceral organs, especially the liver [47]. In a clinical context, it has been observed that patients exhibiting metastases in visceral sites upon progression tend to manifest elevated levels of ctDNA alongside higher detection rates, as opposed to those whose progression involves metastases in the lymph nodes, subcutaneous tissue, or pulmonary lesions. Marsavela et al. reported that among cases progressing with metastases in cutaneous, subcutaneous, or nodal locations, only 9 out of 19 exhibited detectable ctDNA. Notably, in patients with exclusive intracranial metastatic involvement, all but one displayed undetectable ctDNA upon progression, in contrast to the heightened detection rates observed in cases with extracranial disease dissemination [48].

This highlights the complexity of ctDNA dynamics and the importance of considering specific disease characteristics when interpreting ctDNA results.

The introduction of PD-1 and/or CTLA-4 immunotherapies marked a revolutionary shift in the clinical management of patients with advanced melanoma. Both are now the standard of care in daily practice. More recently, the new LAG-3 antibody relatlimab combined with nivolumab showed event better data vs. nivolumab alone [42]. Additional immune checkpoint inhibitor (ICI) combinations are currently undergoing early-phase clinical trials. Consequently, a pivotal focus in the future will be on optimizing the sequencing of these therapies to extend PFS and OS. CtDNA may emerge as a tool that could facilitate more informed decisions regarding therapy switch. Addressing the heterogeneity of response to ICI represents a significant challenge. Identifying predictive biomarkers is crucial for tailoring treatment to the individual. Several studies have indicated that ctDNA can serve as a baseline biomarker for predicting responses to ICIs.

In a study comprising 141 melanoma patients having been treated with immunotherapy, notable results emerged. Those with undetectable ctDNA at the beginning, in contrast to individuals with detectable ctDNA levels determined through a real-time polymerase chain reaction (RT-PCR) for BRAF or a droplet digital polymerase chain reaction (ddPCR) for NRAS in 1 mL of plasma, demonstrated a PFS of 26 weeks compared to 9 weeks (HR 0.47; *p* = 0.01).

Additionally, OS for the undetectable ctDNA group treated with immunotherapy was not reached, while it was 21.3 weeks for the detectable ctDNA group (HR = 0.37; *p* = 0.005). These results underscore the potential of ctDNA as a predictive biomarker for response to pembrolizumab treatment in this cohort [49].

ctDNA has demonstrated its utility as an on-treatment biomarker for assessing responses to ICIs. Initially, studies revealed that increasing ctDNA levels during longitudinal sampling of patients undergoing ICI treatment correlated with disease progression. This suggests that monitoring ctDNA dynamics during treatment could provide valuable insights into the therapeutic response and disease status in individuals receiving ICIs [50]. ctDNA could serve as a valuable tool in elucidating response during pseudo-progression, a phenomenon defined by an initial increase in the size of the primary tumor or the emergence of a new lesion followed by subsequent tumor regression. In around 10–30% of melanoma patients undergoing ICI treatment, pseudo-progression is observed. A positive circulating tumor ctDNA profile, either undetectable initially or transitioning to undetectable by week 12, proved effective in distinguishing genuine progression from pseudo-progression. This method showed a sensitivity of 90% (95% CI: 68–99%) and a specificity of 100% (95% CI: 66–100%) [51].

In a retrospective analysis involving 555 plasma samples from 69 advanced melanoma patients, a personalized ctDNA assay was employed. The study was divided into three cohorts, demonstrating that molecular residual disease (MRD) positivity in stage III patients receiving adjuvant ICI therapy (cohort A) correlated with significantly shorter distant metastasis-free survival. Moreover, increasing ctDNA levels post-surgery or pre-treatment to 6 weeks after ICIs predicted shorter distant metastasis-free survival in cohort A and shorter progression-free survival in cohort B. Notably, in cohort C, ctDNA-negative patients remained progression-free, while ctDNA-positive patients experienced disease progression during a median follow-up of 14.67 months [52].

These findings underscore the potential of ctDNA alone or in complex with other biomarkers, providing more accurate assessments of treatment response, especially in scenarios with complex dynamics like pseudo-progression.

In a recent validation study that included patients from the COMBI-d and COMBI-MB trials, which involved the use of dabrafenib plus trametinib, the confirmation of baseline detectable ctDNA as a predictor for PFS and OS during targeted therapy was strengthened. By establishing a cut-off of 64 copies of ctDNA per milliliter through a droplet digital polymerase chain reaction (ddPCR), patients were categorized into low- and high-risk groups. Patients from the low-risk group showed notably longer PFS at 12.7 vs. 6.5 months (HR 1.74; 95% CI: 1.37–2.21, *p* < 0.0001) and OS at 35.1 vs. 13.4 months (HR 2.23; 95% CI: 1.73–2.87, *p* < 0.0001) [38]. These findings underscore the potential of baseline ctDNA levels as a robust predictor of treatment outcomes in patients receiving targeted therapy.

There is evidence suggesting that ctDNA can be employed to monitor patients undergoing targeted therapy. Notably, changes in ctDNA levels may be detectable earlier than alterations in radiological imaging or biochemical markers like LDH. This early detection capability serves as a valuable indicator of treatment efficacy or an early warning sign of potential disease progression. Monitoring ctDNA provides a dynamic and potentially more responsive approach to assessing treatment response compared to traditional markers [53] (Table 1).

## 5. Mutations in ctDNA and Their Significance for Melanoma Prognosis

Melanoma is genomically complex, and compared to other tumor types, melanoma exhibits a relatively high number of mutations, particularly in tumors linked to sun exposure as an etiological factor [54]. In melanoma, the heightened interest in ctDNA stems from its dual characteristics of possessing a high mutational burden and the early emergence of somatic mutations in key driver genes during tumorigenesis in a substantial portion of cases. This distinctive combination establishes an ideal scenario, positioning mutant ctDNA as a valuable biomarker for both prognosis and ongoing monitoring in melanoma patients [38].

In the meta-analysis conducted by Yang Zheng and colleagues on the significance of ctDNA mutations in melanoma, their findings indicated a notable association between ctDNA mutations and the prognosis of melanoma patients. Specifically, patients with detectable ctDNA mutations showed a tendency towards unfavorable OS compared to those where ctDNA mutations were not detected, whether at the baseline or after treatment. Moreover, individuals with low or undetectable ctDNA mutations at the baseline exhibited better progression-free survival (PFS) in contrast to those with high ctDNA mutations, and the presence of detectable ctDNA at baseline was linked to adverse PFS. Additionally, patients with decreasing ctDNA levels demonstrated a trend towards favorable PFS compared to those with increasing ctDNA levels. Notably, the presence of ctDNA BRAFV600 mutations emerged as a prognostic biomarker with similar predictive value. Specifically, patients with detectable BRAFV600 ctDNA at baseline tended to experience worse OS compared to those with undetectable BRAFV600 ctDNA, and baseline detectable BRAFV600 ctDNA was associated with poorer PFS [55].

NRAS mutations are observed in over 20% of individuals with cutaneous melanoma, leading to the activation of various cellular signaling pathways, including MAPK and PI3K. This activation contributes to processes such as cell growth, proliferation, and cell cycle deregulation. The TCGA network, employing whole exome sequence examination in patients diagnosed with local and/or advanced melanoma, recognized four unique genomic subtypes: those with mutations in NF1, mutations in BRAF, mutations in NRAS, and those categorized as triple wild type [56].

The classification into these genomic subtypes may hold predictive value, especially concerning the available therapeutic targets. Consequently, employing multigene ctDNA mutation detection can enhance the accuracy of predicting the prognosis of individuals with melanoma, providing valuable insights into potential therapeutic strategies [57].

In melanoma, either BRAFV600 or NRASQ61 hotspot mutations are typically found mutually exclusively in approximately two-thirds of metastatic tumors. Specifically, patients with BRAFV600-mutant tumors can undergo highly effective treatment with BRAF plus MEK-targeted therapies [58].

Several studies have demonstrated that ctDNA can serve as a predictive biomarker of response to targeted therapy when assessed at baseline. Additionally, ctDNA has proven valuable as an on-treatment biomarker, aiding in monitoring response and detecting disease progression. Furthermore, ctDNA functions as a tool to identify mechanisms of resistance, offering insights into the factors contributing to treatment resistance in various therapeutic contexts. The initial level of mutant BRAF in cfDNA has been demonstrated as a predictive biomarker for the duration of therapy in patients undergoing treatment with BRAF/MEK inhibitors [53]. This suggests that the baseline presence of mutant BRAF in ctDNA can provide insights into the expected duration of the therapeutic response in individuals receiving BRAF/MEK inhibitor treatment. The presence of mutant BRAF copies in ctDNA was linked to lower response rates in comparison to patients with undetectable mutant BRAF. Additionally, individuals with detectable mutant BRAF in ctDNA experienced shorter OS and PFS than those with undetectable mutant BRAF [59].

In the last few years, researchers have investigated tumor mutation burden (TMB) as a genetic indicator for forecasting how patients with melanoma will respond to ICI treatments. This interest arises from conflicting findings in studies regarding the correlation between elevated TMB and the OS benefits of ICI. An elevated TMB is linked to enhanced OS and PFS among melanoma patients undergoing treatment with ICI monotherapy [60].

In a smaller-scale study focused on ctDNA panels, the findings suggested that TERT (telomerase reverse transcriptase) was the most frequently mutated gene identified in ctDNA, and its presence was associated with an unfavorable prognosis. Additionally, the study revealed a higher concentration of ctDNA in patients with a high metastatic load. This observation implies that more aggressive tumors release an increased amount of ctDNA into the bloodstream. These results underscore the potential of ctDNA, especially mutations in genes like *TERT*, as informative indicators of disease aggressiveness and prognostic factors in the context of cancer [61].

Certainly, numerous studies have emphasized the ability of ctDNA to identify resistance pathways to systemic treatment. These inquiries have illustrated the occurrence of mutations in genes like *PIK3A*, *NRAS*, *MAP2K1*, and *AKT1* among patients with melanoma undergoing targeted therapy. These genetic alterations are acknowledged as resistance mechanisms to MAPK-targeted therapy in patients with melanoma and have been associated with subsequent progressive disease detected by radiological imaging, such as CT scans [38].

## 6. Conclusions

This review underscores the profound and transformative impact of ctDNA in reshaping the landscape of clinical decisions for advanced melanoma. The revolutionary advancements brought about by anti-PD1 and anti-CTLA-4 immunotherapies and targeted therapies have undeniably elevated the standard of patient care, yet the need for precise and dynamic biomarkers persists. ctDNA emerges as a promising blood-based tool, offering insights into prognosis, treatment response, and resistance mechanisms associated with advanced melanoma.

The technical nuances of detecting ctDNA highlight the importance of meticulous sample handling, with plasma identified as the optimal substrate for conducting analysis. The presented meta-analyses add weight to the narrative by revealing a robust association between ctDNA mutations and melanoma prognosis, providing clinicians with valuable predictive and prognostic information. The exploration of NRAS mutations and genomic subtypes enhances the understanding of melanoma heterogeneity, guiding the implementation of multigene ctDNA mutation detection for refined prognostication.

In the treatment area, ctDNA proves instrumental in guiding therapeutic decisions for both immunotherapy and targeted therapy. Its role as a baseline biomarker for immunotherapy response prediction, as well as an on-treatment monitor and a discriminator of true progression from pseudo-progression, highlights its multifaceted utility. The predictive power of ctDNA in targeted therapy, especially its ability to identify mechanisms of resistance, positions it as a valuable asset in tailoring treatment strategies for enhanced efficacy.

The journey towards precision medicine in melanoma care is increasingly reliant on the dynamic nature of ctDNA. Its early detection capabilities, real-time monitoring, and ability to elucidate resistance mechanisms mark a paradigm shift in clinical decision-making. As the field continues to evolve, ctDNA stands out as a beacon for personalized and informed clinical decisions, offering a promising trajectory for enhancing patient outcomes in the area of advanced melanoma (Figure 2).

## Figures and Tables

**Figure 1 cancers-16-01197-f001:**
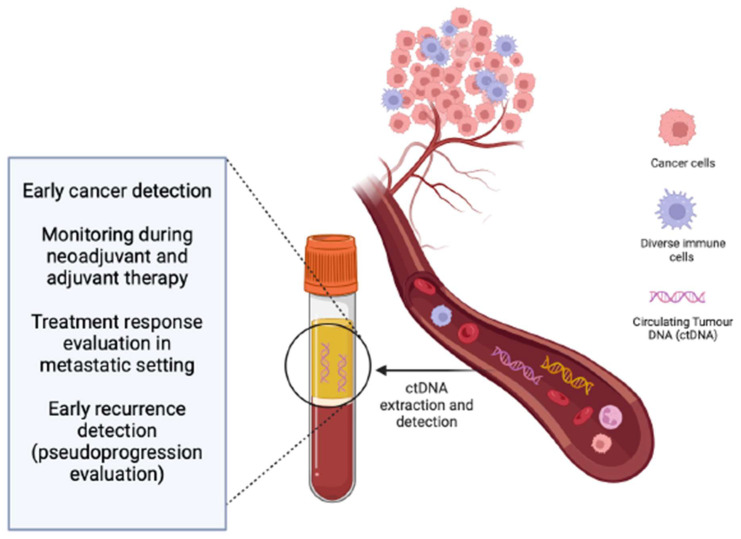
Potential clinical utility of ctDNA.

**Figure 2 cancers-16-01197-f002:**
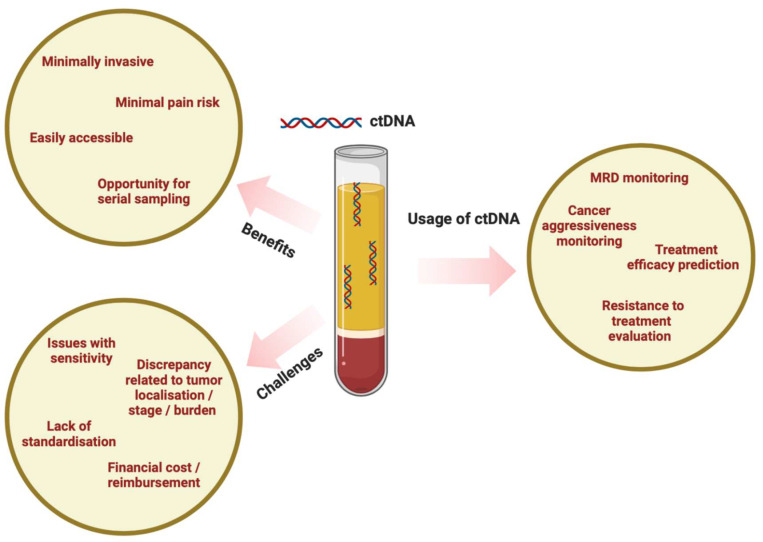
ctDNA in melanoma management.

**Table 1 cancers-16-01197-t001:** Compilation of articles on the utility of ctDNA in melanoma.

ctDNA in Locally Advanced Melanoma	ctDNA in Metastatic Melanoma
Cheng Y et al. (2015) [36]. Gandini S. et al. (2021) [37]. Lee JH et al. (2019) [38]. Tan L. et al. (2019) [39]. Forschner A. et al. (2022) [40]. Huang N. et al. (2022) [42].	Wong SQ. et al. (2017) [41]. Chang-Hao TS et al. (2015) [43]. Santiago-Walker A. et al. (2016) [44]. Lipson EJ. et al. (2014) [45]. Tawbi HA. et al. (2022) [46]. Marsavela G. et al. (2020) [47]. Lee JH. et al. (2018) [48]. Eroglu Z. et al. (2023) [49]. Chang GA. et al. (2015) [50].

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
