# Peer review of "Harnessing ctDNA in Advanced Melanoma: A Promising Tool for Informed Clinical Decisions"

_cancers, 2024, doi:10.3390/cancers16061197_

Round 1
Reviewer 1 Report
Comments and Suggestions for Authors
This condensed systematic review addresses the potential clinical implications of ctDNA as a biomarker in cutaneous melanoma. The article is well written and highlights several areas of clinical utility which is of major interest.
Information on how the literature review was done, i.e. how scientific papers were identified and included should be added to the ms with inclusion and exclusion criteria.
Specific comments
Introduction, although biomarkers in blood is limited several institutions are using also S100B as clinical biomarker. That’s should preferably be added.
Also in the Introduction, an overall information of liquid biopsies in relation to ctDNA, circulating tumor cells, exosomes etc could be of value to comment to put ctDNA in a context in the “family of liquid biopsies”.
Several areas of clinical utility are mentioned, one that is not mentioned is the use of ctDNA diagnostically when biopsy is not feasible. Please comment and add as suitable.
Considering the short half-life of cfDNA, in the ms its stated “to be estimated at 1 to 2 hours”. What implications does this have on blood sampling? Does it exist a circadian rhythm of ctDNA levels or similar? What routines for ctDNA assessment exist?
Other sources of ctDNA than originating from the tumor the cutaneous melanoma is not brought up. That should preferably be added. What’s the relation to other skin cancers or benign skin lesions like naevi where BRAF mutations also exist? Do these tumors/lesions also shed/exert ctDNA? What is known? Is this a source of ctDNA that interferes?
Correlation to tumor burden and site of metastases is discussed briefly, this paragraph should preferably be expanded for a more detailed on the understanding related to eg, expected lower levels of ctDNA in patients with brain or subcutaneous metastases only vs presence of visceral metastases.
I would suggest adding a short summary with bullet points in a separate box as a separate figure. “Factors to consider for clinical use of ctDNA in cutaneous melanoma” or similar. That would help the reader to quickly grasp the context of the ms.
Comments on the Quality of English LanguageGood.
Author Response
Thank you very much for your comments and insights.
Introduction, although biomarkers in blood is limited several institutions are using also S100B as clinical biomarker. That’s should preferably be added.
Answer: Please see line 59.
Also in the Introduction, an overall information of liquid biopsies in relation to ctDNA, circulating tumor cells, exosomes etc could be of value to comment to put ctDNA in a context in the “family of liquid biopsies”.
Answer: The focus of this manuscript was exclusively directed towards the examination of DNA present in blood and exploring its potential applications.
Several areas of clinical utility are mentioned, one that is not mentioned is the use of ctDNA diagnostically when biopsy is not feasible. Please comment and add as suitable.
Answer: Please see lines 144-147.
Considering the short half-life of cfDNA, in the ms its stated “to be estimated at 1 to 2 hours”. What implications does this have on blood sampling? Does it exist a circadian rhythm of ctDNA levels or similar? What routines for ctDNA assessment exist?
Answer: There is no any data regarding circadian rhythm influence on ctDNA levels in melanoma patients. Poulet et al (Scientific Reports volume 13, article number: 21675 (2023)) assessed how the circadian rhythm affects the release of cell-free DNA in healthy individuals over a full day. Ten healthy women had blood samples taken at 8 am, while 20 healthy men had multiple blood samples taken at various times throughout the day: 8 am, 9 am, 12 pm, 4 pm, 8 pm, 12 am (midnight), 4 am (next day), and 8 am (next day). The researchers utilized digital droplet-based PCR for their analysis. In conclusion, any influence of circadian rhythm was not detected. Currently, there is no standardized protocol for the assessment of circulating tumor DNA. The field is still evolving, with various methodologies and techniques being utilized by different research groups and clinical laboratories. This lack of standardization poses challenges in comparing results across studies and implementing ctDNA analysis consistently in clinical practice.
Other sources of ctDNA than originating from the tumor the cutaneous melanoma is not brought up. That should preferably be added. What’s the relation to other skin cancers or benign skin lesions like naevi where BRAF mutations also exist? Do these tumors/lesions also shed/exert ctDNA?What is known? Is this a source of ctDNA that interferes?
Answer: The idea was to focus on circulating tumor DNA specifically within the context of melanoma, rather than benign skin lesions or other non-melanoma skin cancers. This narrowed focus allows for a more targeted evaluation of the role and utility of ctDNA in melanoma diagnosis, prognosis, treatment response monitoring, and disease surveillance. By excluding benign skin lesions and non-melanoma skin cancers from the discussion, the aim was to delve deeper into the challenges and opportunities associated with ctDNA analysis in the context of melanoma.
Correlation to tumor burden and site of metastases is discussed briefly, this paragraph should preferably be expanded for a more detailed on the understanding related to eg, expected lower levels of ctDNA in patients with brain or subcutaneous metastases only vs presence of visceral metastases.
Answer: Please see lines: 226-241.
I would suggest adding a short summary with bullet points in a separate box as a separate figure.
“Factors to consider for clinical use of ctDNA in cutaneous melanoma” or similar. That would help the reader to quickly grasp the context of the ms.
Answer: Please see Figure 1.
Reviewer 2 Report
Comments and Suggestions for Authors
This is an important review about ctDNA in advanced melanoma. The authors summarizes significance of circulating tumor DNA (ctDNA) as a non-invasive 'real-time' biomarker, which is capable to offer diagnostic and prognostic insights prior to therapy, during the course of treatment, and upon disease progression.
The references are up-to-date.
Comments:
The affiliation of the authors are not complete; name of the departments and the cities and countries are missing from the first page.
The references are always correct:
etc Ref 1: is not about the incidence of cutaneous melanoma …
Ref3: is about comparative genomics of melanoma I did not find anything about the the adoption of immunotherapies targeting PD1 and/or CTLA-4 …
Ref 4: is about: Outcomes after progression of disease with anti-PD-1/PDL1 therapy for advanced melanoma and not about …Imaging scans…
Ref 5: is not about resistance and toxicity but: . Circulating tumour DNA and melanoma survival …
Some recent references are missing Line 78-81.
Author Response
Thank you very much for your comments and insights.
- The affiliation of the authors are not complete; name of the departments and the cities and countries are missing from the first page.
Answer: the author affiliations were updated.
- References were updated to more accurate ones:
- Reference [1] updated to: Arnold M, Singh D, Laversanne M, Vignat J, Vaccarella S, Meheus F, Cust AE, Vries ED, Whiteman DC, Bray F. Global burden of cutaneous melanoma in 2020 and projections to 2040. JAMA Dermatol. 2022;158(5):495-503. doi:10.1001/jamadermatol.2022.0160.
- Reference [3] updated to: Adams R, Coumbe JEM, Coumbe BGT, Thomas J, Willsmore Z, Dimitrievska M, Yasuzawa-Parker M, Hoyle M, Ingar S, Geh JLC et al. BRAF inhibitors and their immunological effects in malignant melanoma. Expert Review of Clinical Immunology. 2022;(4):347-362. doi:10.1080/1744666x.2022.2044796.
- Refecence [4] updated to: Aide N, Iravani A, Prigent K, Kottler D, Alipour R, Hicks RJ. PET/CT variants and pitfalls in malignant melanoma. Cancer imaging. 2022Jan4;22(1):3. doi:10.1186/s40644-021-00440-4.
- Reference [5] updated to: Patel M, Eckburg A, Gantiwala S, Hart Z, Dein J, Lam K, Puri N. Resistance to molecularly targeted therapies in melanoma. Cancers (Basel). 2021 Mar; 13(5):1115. doi:10.3390/cancers13051115.
- Some recent references are missing Line 78-81.
Answer: Reference to line 78 was added.
Reviewer 3 Report
Comments and Suggestions for Authors
The manuscript by Pikturniere et al. is a very interesting review regarding the potential usage of ctDNA on clinical decision in patients with malignant melanoma, prior, during and after treatment.
The article is very well written, but there are some minor issues to be taken into consideration.
1. Please provide a better description of the Author affiliations.
2. Line 186-187 the reference is missing.
3. Line 242-243 something is missing. The sentence does not make sense. Do the authors mean that the mean OS was not reached?
4. Overall Tables are missing. I suggest that the authors include one or more tables summarising the presented articles in order to make it easier to follow the provided information, maybe subdividing the information into prior/during/or after treatment.
Author Response
Thank you very much for your comments and insights.
- Please provide a better description of the Author affiliations
Answer: author affiliations were updated.
- Line 186-187 the reference is missing.
Answer: reference to line 196 was added [38].
- Line 242-243 something is missing. The sentence does not make sense. Do the authors mean that the mean OS was not reached?
Answer: lines 261-264: yes, OS was not reached in undetectable ctDNA group because they tend to have better prognosis. I updated the sentence to make it clearer:
‘Additionally, OS for the undetectable ctDNA group treated with immunotherapy was not reached, while it was 21.3 weeks for the detectable ctDNA group (HR = 0.37; P = 0.005). These results underscore the potential of ctDNA as a predictive biomarker for response to pembrolizumab treatment in this cohort [49].’
- Overall Tables are missing. I suggest that the authors include one or more tables summarising the presented articles in order to make it easier to follow the provided information, maybe subdividing the information into prior/during/or after treatment.
Answer: we added the table to the article: Table 1. Compilation of articles on the utility of ctDNA in melanoma.